# Review on Retrospective Procedures to Correct Retinal Motion Artefacts in OCT Imaging

**Luisa Sánchez Brea** [1,†] 🔟, **Danilo Andrade De Jesus** [1,†] 🔟, **Muhammad Faizan Shirazi** [2] 🔟, **Michael Pircher** [2] 🔟, **Theo van Walsum** [1] and **Stefan Klein** [1,*] 🔟

[1]   Department of Radiology and Nuclear Medicine, Biomedical Imaging Group Rotterdam, Erasmus MC, 3015GD Rotterdam, The Netherlands
[2]   Center for Medical Physics and Biomedical Engineering, Medical University of Vienna, 1090 Vienna, Austria
*   Correspondence: s.klein@erasmusmc.nl
†   These authors contributed equally to this work.

**Abstract:** Motion artefacts from involuntary changes in eye fixation remain a major imaging issue in optical coherence tomography (OCT). This paper reviews the state-of-the-art of retrospective procedures to correct retinal motion and axial eye motion artefacts in OCT imaging. Following an overview of motion induced artefacts and correction strategies, a chronological survey of retrospective approaches since the introduction of OCT until the current days is presented. Pre-processing, registration, and validation techniques are described. The review finishes by discussing the limitations of the current techniques and the challenges to be tackled in future developments.

**Keywords:** optical coherence tomography; retinal motion artefacts; image registration

## 1. Introduction

Optical coherence tomography (OCT) was introduced as a new modality for cross-sectional imaging of biological tissue in 1991 [1] and, since then, it has become one of the most successful technologies in the history of biomedical optics [2]. OCT is nowadays used in a wide range of research fields, but it is in ophthalmic imaging where it finds its main application [3,4]. Retinal imaging diagnostics has been completely revolutionized over the last decade by OCT, which has become the gold standard imaging modality [5,6].

OCT is an interferometric technique that detects the magnitude of reflected or backscattered light from different depths. An OCT system probes the sample with low a temporal coherence light beam, letting the backscattered light interfere with a reference beam from the same light source. From the resulting interference signal, the reflectivity profile of the sample along the beam axis (A-scan) is derived. Cross-sectional images (B-scans) are generated by recording of multiple A-scans, while scanning the incident light beam transversely over the sample. Three-dimensional (3D) volumetric data are generated by acquiring sequential cross-sectional images in a raster pattern [4,5].

The time needed for acquiring each dataset depends mainly on its spatial dimension and the imaging speed of the device, which is broadly given by the A-scan acquisition rate. Typical state-of-the-art research systems feature A-scan rates of up to several hundreds of kilohertz [7–9], usually limited by the camera read-out rate in Spectral Domain-OCT (SD-OCT), or the sweep rate of the light source in Swept Source-OCT (SS-OCT) [10]. In older systems that are based on Time Domain OCT (TD-OCT), the maximum velocity of the reference arm length change limited the A-scan rate which typically was below 2 kHz. Fast image acquisition in vivo is very important for sampling correctly large field-of-view (LFoV) OCT volumes, as retinal motion cannot be avoided. Even when the subject fixates on a stationary object, the eye carries out involuntary small and

rapid movements that may reach several arcminutes with a frequency between 10 and 100 Hz [11,12]. These involuntary fixational eye movements are usually categorized as high frequency tremors, rapid microsaccades, or slow drifts, depending on their frequency and magnitude [11,12]. In addition, axial eye motion (introduced for example by heart beat) may distort B-scan images and result in deformed 3D data of the retina.

The degree of inter- and intra-frame OCT image distortion generated by involuntary eye movements, as well as potential signal attenuation (due to the fringe wash out in the case of SD-OCT), increases with the acquisition time. This compromises the detection and tracking of the progression of retinal diseases. Consequently, complementary retinal tracking hardware or post-processing software procedures are often used to correct the distorted data.

A substantial number of studies using LFoV OCT volumes are published on a monthly basis. However, only a minority tackle motion correction, even though a minimal distortion can easily bias structural analysis, such as measurements of retinal layer thicknesses. Although some correction methods have been published over the last years, motion artefacts from axial movement, changes in fixation, and saccades remain a major problem in OCT imaging [13,14].

To the best of our knowledge, only Baghaie et al. in 2015 [15] and later in 2017 [16] provided a review on registration methodologies used in retinal OCT imaging. A succinct overview on registration approaches for speckle noise reduction, multimodality imaging, and motion correction is given in 2015, whereas, in 2017, an extended review comparing hardware-based versus software-based techniques is provided. While hardware approaches are well presented, only a few publications on software-based approaches were covered. Moreover, little information was given on validation techniques, which are one of the major concerns on OCT registration due to the lack of a ground truth for the retinal structure.

In this work, a broader and extensive review focused on retrospective procedures to correct retinal motion, covering 38 publications, is provided. The pre-processing, registration, and validation techniques used in these publications, whenever provided, are described and discussed. The remainder of the paper is organized as follows. In Section 2, the retinal motion that affects OCT imaging is explained. Section 3 describes the literature search methodology and summarizes the topics that are tackled in the survey, namely the motion correction approaches, pre-processing, and validation techniques. The retrospective procedures used to correct retinal motion in OCT volumes are reviewed in Section 4. Section 5 provides a discussion on the reviewed approaches, followed by the conclusions in Section 6.

## 2. Motion and Distortion Induced Artefacts

The direction in which the B-scans are acquired is also known as fast direction and, in general, it corresponds to horizontal cross-sections of the retina, as depicted in Figure 1.The direction orthogonal to the fast direction is named the slow scanning direction. The coronal or en face plane is spanned by these two directions.

An OCT volume can eventually be distorted by several factors, including the geometry of the OCT optical setup and motion of the eye. In this review, only eye motion correction is extensively covered, although a non-exhaustive overview on optical distortions that may arise within OCT scanning of the retina is given in the following subsection.

### 2.1. Optical Distortions

Optical distortions in OCT imaging result from light rays that deviate from the optimal path. This may happen due to aberrations in the optics interface between the scanner and the retina, as well as misalignment or other limitations of the optical setup [17–20]. These distortions may be caused by non-telecentric scan geometries, refraction of the probing light in the sample or, in the case of time domain systems, nonlinearities in the reference arm scan. To correct these artefacts, a few approaches have been proposed. Westphal et al. [17] presented a methodology for correcting non-telecentric scan patterns, as well as an approach for refraction correction in layered media based on Fermat's principle.

Podoleanu et al. [18] quantified the distortions introduced by scanning and eye motion using two quantities, a lateral and an axial distortion. These values represent deviations of the image points from the points where the user imagines or expects the points to be. More recently, Zawadzki et al. [19] suggested to correct the optical distortions by incorporating the geometry of the scanning beam as part of a custom volume rendering software.

Although a few software-based approaches have been suggested to correct optical distortions induced by the setup (OCT device), artefacts caused by retinal motion or varying subject alignment in respect to the OCT system remain and will influence the outcome of longitudinal studies. Thus, intra- and inter-subject OCT data will be affected, hampering subsequent interpretation and analysis.

## 2.2. Involuntary Eye Motion

Involuntary movements of the eye, or the retina in particular, can lead to significant distortions within the OCT volumetric data acquisition [21] which, in some cases, are easily seen in the raw data. An example of a distorted OCT volume is shown in Figure 1. The represented planes can be taken as reference for the direction nomenclature used in this review. In this unregistered OCT volume, acquired with a custom made device at Medical University of Vienna (SS-OCT at 1050 nm with 200 kHz A-scan rate, 200 Hz B-scan rate) [22], the distortions are visible in the slow and coronal (en face) planes. The fast plane is often considered artefact free, as the frame rate acquisition in modern devices is faster than the motion expected to be observed in the retina. Nevertheless, axial motion, respiration, vascular pulsation, and involuntary fixational eye movements [11–13], described in the following subsections, may all contribute to coronal and axial distortions of the OCT data.

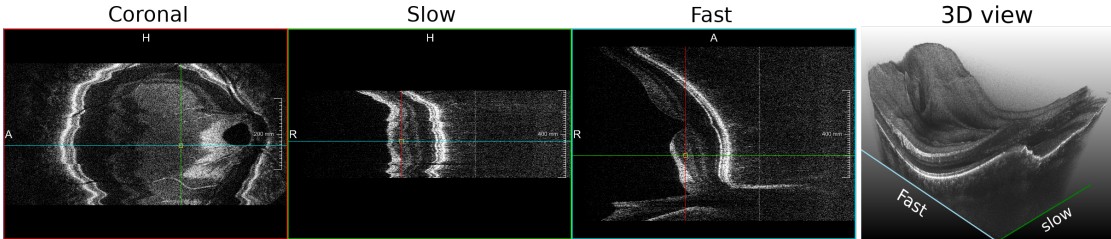

**Figure 1.** Planes Coronal, Slow, Fast, and 3D view of a raw unregistered OCT volume acquired with a custom made device at Medical University of Vienna. Scanning angles are $45°$ on the $x$-direction and $25°$ on the $y$-direction, where $1°$ corresponds approximately to 300 μm. The depth extension is 471 pixels (3.3 μm pixel$^{-1}$). Only the fast plane is visually artefact free, since the acquisition rate was 200k A-scans s$^{-1}$ resulting in 200 B-scans per second.

### 2.2.1. Tremors

Among all motion sources, tremors have the least weight on OCT volumetric distortion, as their amplitudes are around $8.5''$, which corresponds to the diameter of a cone photoreceptor (approximately 6 μm [23]). Taking into account that, on average, the lateral optical resolution of commercial OCT is approximately between 15–20 μm, we would expect the amplitude of the tremors to fall within the range of the recording system noise, even though their frequency may range between 25–90 Hz [24]. Nevertheless, motion caused by tremors may become an issue on future OCT configurations such as in Adaptive Optics OCT (AO-OCT), where the lateral resolution is approximately 2–5 μm [25,26].

### 2.2.2. Drifts

Drifts are slow retinal movements that can move the image of a fixated object on the retina across a dozen photoreceptors. Yarbus [27] was one of the first to report a drift mean speed, $6's^{-1}$, and a maximum speed of $30s^{-1}$. Years later, Srebo et al. [28] reported a mean drift speed of $24.6's^{-1}$. In both studies, it was observed that drifts occur simultaneously with tremors and between microsaccades.

However, their amplitude (between 0.8′ and 31.4′ [12,23]) is much larger than in tremors, and therefore drifts are likely to affect 3D-OCT data.

### 2.2.3. Microsaccades

Microsaccades are the fastest and largest involuntary eye movements that occur during fixation, carrying the image of a fixated object on the retina across a large number of photoreceptors, up to several hundreds [12,23]. Different speeds of this motion have been reported. Schulz [29] and Riggs et al. [30] observed a maximum speed between 4.2–55° s$^{-1}$, whereas more recently, in 2002, Moller et al. [31] measured higher values (28–97° s$^{-1}$). The frequency, amplitude, and direction of the microssacades have been linked to displacements of the image across the photoreceptors. For example, if drifts carry the image of the fixation target on the retina away from the fovea, microsaccades will likely correct the displacement, as depicted in Figure 2. However, non-corrective microsaccades may also occur [23].

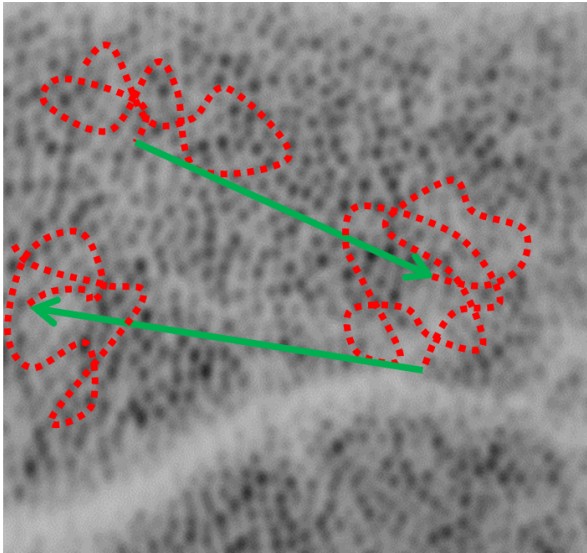

**Figure 2.** Schematic of the involuntary eye movements drawn on an in vivo AO-OCT en face image showing individual cone photoreceptors (width ∼1–1.2 µm [32]), expected to affect the OCT image quality. Slow drifts are represented by the dotted lines, whereas the fast microsaccades are shown by the green arrows. The black dots represent individual cone photoreceptors. The white curved line is caused by shadowing of anterior retinal vessels. The distance between the cones at the specific eccentricity from the fovea (∼4°) is approximately 3–4 µm .

### 2.2.4. Pulsing and Respiration

Retinal blood pulsation has also been reported as a possible cause of OCT imaging distortion. Robinson et al. [33] reported that, in the case of imaging retinal blood vessels, each acquisition is associated with a unique position of the blood vessels compared to the background tissues. Kinkelder et al. [34], following the same research line, reported that the heartbeat causes an arterial blood pressure pulse in the arteria carotis, inducing a head movement (average peak-to-peak amplitude of 81 µm $\pm$ 3.5 µm per second) that is translated to the eye and it is visible during the retinal OCT scanning. Moreover, it is mentioned in their work that respiration also causes a slow and small amplitude movement of the head, with a frequency of 0.2 Hz. Both motions clearly affect volumetric OCT data but may in addition cause distortions of B-scans.

## 3. Methods

This section presents the literature search strategy, an overview of the structure of the survey and the motion correction approaches, as well as the most common pre-processing and validation techniques used in retroprospective OCT retinal motion correction.

### 3.1. Literature Search

The literature search was carried out using Google Scholar and Web of Science. A query was designed, combining different keywords that defined the problem (image registration, movement correction, motion correction, motion artifacts) and the imaging modality (optical coherence tomography, OCT) to be tackled in this review. The anatomical area of interest was also taken into account (retina, retinal). For each search result, the eligibility criteria were applied. Then, within each selected work, references were checked, and the eligibility criteria were applied to each of them, continuing the iterative process. No restrictions were made with respect to the publication date. Regarding the type of publication, only the patents were excluded.

#### 3.1.1. Inclusion Criteria

Only original research published in English was included. Studies with the main focus on registration or other imaging processing techniques applied to OCT data were included, as far as enough information being provided on the registration procedure. Research that used registration as pre-processing was included if some information on the basics of the registration was provided (e.g., corrected directions and type of transformations). Approaches focused on widening the field of view by mosaicing were also included, as long as the methodology was relevant for OCT intra- or inter-volume registration and motion correction.

#### 3.1.2. Exclusion Criteria

Hardware based approaches, as well as eye tracking studies, and theoretical procedures were excluded, as they were out of the scope of this review. Algorithms focused on OCT-Angiography (OCT-A), AO-OCT or other OCT modalities rather than standard OCT imaging were excluded. Research focused on multi-modality registration was also excluded, except for those that included intra- or inter-volume OCT registration and motion correction. Lastly, studies mentioning registration without enough details to understand the intrinsic process were also not included.

### 3.2. Overview

The spectrum of possibilities on motion correction is vast, and different options for classifying these approaches could be considered. In this review, a division taking into account the moment when the correction is applied was initially considered: simultaneously with the acquisition (prospective) or after the acquisition (retrospective). Since the review is focused on the latter, two aspects were taken into account to further divide this category: the planes handled to perform the registration (fast, slow, coronal) and the input data available (single, complementary, and multi-modality acquisitions). Lastly, using the different aspects of the registration approaches, and focusing on related tasks, such as preprocessing and validation techniques, the retrospective registration procedures are extensively reviewed in chronological order: since the introduction of OCT in 1991, until the end of 2018.

### 3.3. Motion Correction Strategies

This section details the different motion correction strategies, following the classification established in this survey. It starts with the prospective and retrospective approaches, further dividing the latter depending on the input data and the planes handled by the registration.

### 3.3.1. Prospective Approaches

A number of modern commercial devices include additional hardware for correcting movement, such as retinal trackers [35], although the measurements generated by these instruments are not interchangeable [36]. A few examples are: Spectralis OCT (Heidelberg Engineering, Heidelberg, Germany), which uses auxiliary Scanning Laser Ophthalmoscopy (SLO) images; RTVue (Optovue Inc., Fremont, CA, USA), which uses a full-field fundus image; or tracking OCT from Physical Sciences Inc. (PSI, Andover, MA, USA), which uses a separate tracking beam for observing motion related reflectance changes of the retina [37]. Custom made hardware procedures based on new instruments or configurations have also been proposed [38–41]. For more information on prospective/hardware-based approaches to correct retinal motion on OCT imaging, we refer to the article by Baghaie et al. [16].

### 3.3.2. Retrospective Approaches

Retrospective approaches have the goal of finding a spatial, one-to-one correspondence between the voxels of two images/volumes [42]. Such a registration is an iterative process that follows a certain optimization criteria until it reaches the desired value for the similarity metric, which measures how similar both images are [43]. There are many different metrics for measuring similarity, and some popular choices are correlation [44] or mutual information [45]. Another important aspect of the registration is the kind of transformations that are allowed during the alignment process, so that the movement can be compensated, but the image is not deformed. Some of the most common transformations are rigid, affine and non-rigid. It also must be noted that, although the registration is commonly defined using one of the images as reference (pairwise registration), there are also approaches that register a set of images to a common space (groupwise registration) [46]. The latter has the main advantage of not biasing the results depending on the reference image. In this review, retrospective methods were classified according to the input data (single, complementary, and multimodal acquisitions) and to the plane (fast, slow, and coronal) used to drive the registration.

### 3.3.3. Input Data

*Single acquisitions*: These acquisitions correspond either to a cross-sectional image or an OCT volume, as shown in Figure 3. The algorithms applied to this input data can be divided into two large groups: approaches that use landmarks of the image (feature-based registration [47]) and methodologies that process all the content in the image (intensity-based registration). To combine both groups is also possible. Feature-based registration depends on the location of the points of interest (e.g., vasculature, vessel intersections or bifurcations, retinal layers, or lesions), which are used as reference to correct the misalignment. While feature-based registration is mainly combined with manual feature annotation, automated algorithms could be found in the literature demonstrating a good performance. Automated intensity-based registration approaches take into account the information of the whole data. Manual intervention is not required, which usually reduces the operational time, although this may vary depending on the specific algorithm.

Generic registration algorithms can be applied to any single OCT B-scan or volume, without using prior knowledge or additional information. Nevertheless, additional data and comprehension of the retinal motion may help in designing an efficient and robust algorithm to correct OCT data misalignment.

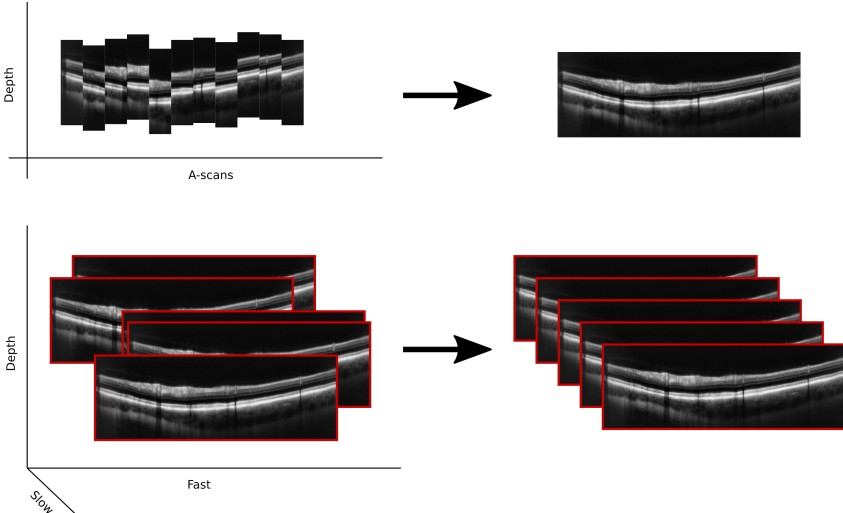

**Figure 3.** Correction of a single cross-section (**top**) and a single volume (**bottom**).

*Complementary acquisitions*: The lack of reliable references in single-modality devices led to the investigation of complementary data acquisition, captured at the same region of interest as the main data. Within this procedure, two alternatives are possible. First, several B-scans or full 3D volumes can be collected at the exact same position and with the same fast plane. Second, an orthogonal raster scan can be acquired (either a few B-scans or a full 3D volume). The latter has gathered more attention by the OCT community over the last years. By having two orthogonal volumes, as in Figure 4, the axial misalignments seen in the slow plane of the reference volume can be corrected by the fast plane of the respective orthogonal volume.

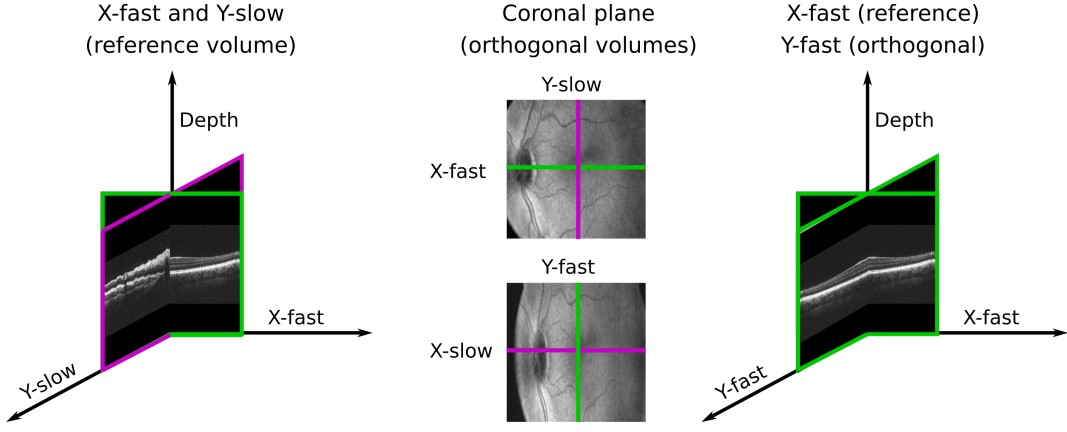

**Figure 4.** Schematic of B-scans acquired orthogonally. The motion artefacts appearing on the slow plane in the reference are corrected using the information of the fast plane of the respective orthogonal volume.

*Multimodal acquisitions*: Registration algorithms are also regularly used for finding an accurate correspondence between modalities that do not exhibit the same characteristics, as depicted in Figure 5. Depending on the acquisition nature, these complementary modalities, such as SLO imaging or Fundus Photography (FP), can be roughly considered motion artefact free when compared to OCT imaging. Hence, a number of studies have opted for using another imaging modality as a reference to correct OCT data. The main advantage of this approach is to have a reference to guide the registration. However, some commonly used modalities, such as SLO, can also be affected by retinal motion themselves [48].

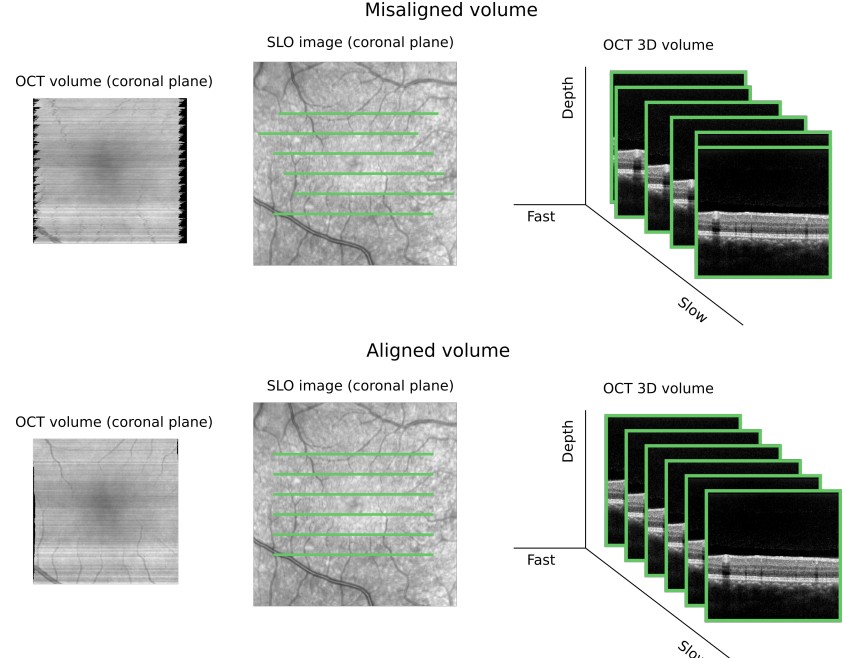

**Figure 5.** Schematic of motion correction based on multimodal acquisition. The upper line denotes a misaligned volume whereas the bottom row shows an aligned volume. The SLO image is used as reference to guide the en face alignment of the OCT volume.

### 3.3.4. Handled Plane

As depicted in Section 2 and, specifically, Figure 1, the planes in the 3D-OCT volume were defined according to the scanning direction. As a consequence of various causes, such as the increase in the acquisition speed or the refinement of the registration algorithms, the planes handled in retinal motion correction change between studies. Thus, a possible classification of the retrospective approaches can be made taking into account the planes used to drive the registration.

*Fast*: The early TD-OCT devices only allowed to acquire a few B-scans within a reasonable exposure time. Thus, the first applied corrections used the information on the fast plane and at the A-scan level ($F_A$). As the acquisition speed increased, the registration methods that used the information on the fast plane switched to correct misalignments between adjacent B-scans ($F_B$).

*Slow*: Motion artifacts are easily observed in the slow plane of an unregistered 3D-OCT volume. A correction can be done by adjusting the smoothness constraints to a wavy profile. This correction can be applied to adjacent A-scans ($S_A$) or B-scans ($S_B$) when orthogonal data are available.

*Coronal*: Lastly, there are some studies [49,50] that have opted for the correction of the fast and slow axis simultaneously, using the coronal plane ($C$). This is commonly linked to multimodal acquisitions, although it is not a necessary condition, as it can be performed by means such as adding smoothness or continuity constraints to the vessels in the en face image.

### 3.4. Pre-Processing Techniques

Some of the most challenging aspects of OCT volumetric data are its low signal-to-noise ratio (SNR), which may compromise the registration performance, and the subtle structural changes in the retinal tissue, usually caused by the early manifestation of pathologies. To mitigate these issues and to ensure the registration is focused on the parts of the image that need to be corrected, pre-processing is often used. Among the reviewed works, two main groups of pre-processing can be considered: image quality enhancement and segmentation of structures of interest.

*Enhancement*: A common pre-processing technique is to smooth the intensity so that the noise and small structural changes are blurred. Filtering, intensity normalization, and background removal have been widely used to enhance the OCT volumes.

*Segmentation*: The most common targets in segmentation are retinal vasculature and retinal layers. For the latter, only a few layers are usually segmented (e.g., retinal boundaries and the retinal pigment epithelium (RPE) layer).

For further details on other pre-processing techniques applied to OCT retinal imaging, we refer to the work by Anantrasirichai et al. [51].

### 3.5. Validation Techniques

Prior to clinical use, the registration techniques employed for correcting OCT misalignments must be validated. In this section, the most common validation approaches used in the reviewed works are presented. The techniques can be grouped in qualitative and quantitative metrics, and approaches that can fall in both groups: simulated data and multimodal comparison.

### 3.5.1. Qualitative Metrics

Visual inspection is the most common evaluation technique used to infer the registration performance. In OCT imaging, a certain smoothness should be expected within the retinal structure, mainly between adjacent A-scans. If an abrupt discontinuity between pixels is observed, one may argue that the registration is not performing well.

### 3.5.2. Quantitative Metrics

An accurate evaluation of the registration performance requires a quantitative and objective analysis. The proposed methods can be roughly divided into techniques that take into account the information available in the whole image, or in specific regions of interest.

*Intensity-based*: Intensity-based metrics quantitatively compare the images using information contained in all voxels. The most commonly used intensity-based metric in OCT registration is mutual information (MI) [45].

*Comparison of annotated structures*: These metrics quantitatively compare certain regions of interest in the image. There are two main approaches: measuring the overlapping of areas or volumes, such as the optic disc; and measuring the distance between structures, such as the edges of a given retinal layer in consecutive B-scans. These techniques may use a manual segmentation as reference. In OCT registration, the most commonly used technique to compute overlapping is the Dice similarity coefficient [52], and the most common choice for distances is the root mean square error (RMSE) with respect to a manual reference.

### 3.5.3. Simulated Data

Simulated or phantom data has also been used to validate the motion correction. Amplitude, frequency and velocity of retinal motion can be simulated in OCT volumes. First, 3D-OCT scans can be deformed by applying a known transformation. Then, the inverse transformation is applied to feature points selected in the deformed volume. Finally, these points are matched with those in the original volume to calculate the voxel distance errors between both images.

### 3.5.4. Multimodality Comparison

Imaging modalities such as FP or SLO have much shorter acquisition times than OCT volumes, and are often considered artifact-free. Thus, they can be used as a reference to infer the registration performance when compared to the OCT en face image. This analysis can be done visually or quantitatively.

*Retinal trackers*: One special case of multimodality comparison is retinal tracking, which uses an additional imaging modality, typically SLO, to guide the OCT acquisition. Although these hardware

approaches are out of the scope of this review, we must mention their usefulness for validation, as they measure the eye motion, effectively providing a reference for the registration.

## 4. Chronological Review of Retrospective Approaches

This section provides an extensive review on retrospective procedures since the introduction of OCT. Tables 1 and 2 summarize the methods and the hardware settings of each OCT device used in the reviewed publications, respectively. The tables are followed by the contents of the reviewed methods, which are presented chronologically and divided in temporal intervals. The temporal intervals were selected taking into account the evolution of the OCT technology and the trends in retinal motion correction. Hence, the first interval represents the period since the introduction of OCT until 2005, when TD-OCT was the main reference, and the motion correction was focused on axial misalignments within cross-sectional images. The predominance of SD-OCT after 2005 and the integration of multimodal imaging in 2009 onwards pushed forward the registration correction between cross-sectional images at the coronal level. By 2011, the combination of multiple data (complementary and multimodal) gained a special interest for improving the motion correction. During these years, there was also a growing trend on the pre-processing importance for improving the registration performance. Algorithms that combined registration and layer segmentation (feature-based) became very common. The last subsection includes publications from 2015 to the present date, when SS-OCT and OCT-A technology started to be used in clinical practice and the A-scan acquisition rate raised up to 100 kHz for commercial systems.

Depending on the research interest, not all the reviewed studies provide all the information listed in Table 1. For instance, regarding the dataset information, some authors specify the number of participants of the study, while others mention number of volumes but do not clarify if the volumes belong to the same individual. Volumes from different individuals are generally preferred for validation, as larger variations between volumes are expected. Thus, Table 1 summarizes the number of volumes from different individuals (healthy and pathological), and the number of volumes as *scans* if no specific information regarding the subjects was provided. Only the studies published after 2005 focused on motion correction with information about the handled plane, the registration method, the validation technique, and the used dataset are described in detail in this section.

**Table 1.** Summary of retrospective procedures to correct retinal motion artefacts in OCT imaging. FP: Fundus Photography, SLO: Scanning Laser Ophthalmoscopy, |: Single image or volume, ‖: Complementary multiple acquisitions, ⊥: Complementary orthogonal acquisitions, −: Not included/mentioned. OCT type described in Table 2 (except those that did not provide enough information on the OCT specifications). Dataset: pathological (P), healthy (H) or (−) not specified.

| Year | Reference | Research Topic | Handled Plane | Method | Preproc. | Validation | OCT Type | Dataset |
|------|-----------|----------------|---------------|--------|----------|------------|----------|---------|
| 1993 | Swanson [53] | OCT retinal imaging: proof of concept | $F_A$ | | | − | − | G | − |
| 1996 | Schuman [54] | Layer thickness repeatability | $F_A$ | | | − | − | G | 10P, 11H |
| 2005 | Ishikawa [55] | Layer segmentation | $F_A$ | | | − | − | A | 24P, 23H |
| 2007 | Zawadzki [19] | Motion correction | $F_B, S_B$ | ⊥ | − | Qualitative | H | 2− |
| | Jorgensen [56] | Image enhanc. | $F_A, F_B$ | | | − | Qualitative | A | − |
| | Fuller [57] | Layer segmentation | $F_B$ | | | − | − | H | − |

**Table 1.** *Cont.*

| Year | Reference | Research Topic | Handled Plane | Method | Preproc. | Validation | OCT Type | Dataset |
|------|-----------|----------------|---------------|--------|----------|------------|----------|---------|
| 2008 | Potsaid [9] | Hardware development | $S_B$ | $\perp$ | Enhanc. | Qualitative | I | – |
|  | Khanifar [58] | Drusen imaging | $F_B$ | $\vert$ | – | Qualitative | J | 31P |
| 2009 | Ricco [59] | Motion correction | $C$ | $\vert$; SLO | Vessel Seg. | Qualitative, Multimodal | SD-OCT | 116P, 48H scans |
|  | Niemeijer1 [60] | Motion correction | $C$ | $\parallel$ | Layer Seg. Enhanc. | Qualitative, Simulated | B | 6H |
|  | Garvin [61] | Layer segmentation | $F_A$ | $\vert$ | Layer Seg. | Qualitative | B | 41H |
|  | Xu [62] | Motion correction | $C$ | $\vert$; SLO | – | Qualitative | B | 24P, 19H |
|  | Tolliver [63] | Motion correction | $S_B$ | $\perp$ | – | Quantitative | B | – |
| 2010 | Robinson [33] | Motion correction | $C$ | $\parallel$ | – | Qualitative | SD-OCT | – |
|  | Kolar [64] | Multimodal registration | $C$ | $\vert$; FP, SLO | Enhanc. Layer Seg. | Qualitative | C | – |
|  | Gibson [65] | 3D-OCT ONH registration | $S_B, C$ | $\parallel$ | ONH Seg. | Qualitative, Quantitative | Custom SD-OCT | – |
|  | Xu [66] | 3D-OCT registration | $F_B, S_A$ | $\vert$ | – | Qualitative | B | – |
| 2011 | Antony [67] | Motion correction | $F_A, S_A$ | $\perp$; FP | Layer Seg. | Qualitative, Multimodal | E | – |
|  | Song [68] | Motion correction | $S_B$ | $\perp$ | Layer Seg. Vessel Seg. | Qualitative, Quantitative | SD-OCT | – |
|  | Li [69] | OCT montage | $C$ | $\parallel$ | RPE Seg. | Quantitative | B | 3P, 3H |
| 2012 | Niemeijer [70] | 3D-OCT registration | $C, F_A, S_A$ | $\parallel$ | Layer Seg. Vessel Seg. | Qualitative, Simulated | F | 5P |
|  | Kraus [71] | Motion correction | $C, F_A, S_A$ | $\perp$ | Enhanc. | Qualitative, Quantitative | K, L | – |
|  | Xu [21] | Registration | $F_B, S_A$ | $\vert$ | – | Qualitative, Simulated | B | 49P, 25H |
| 2013 | He [72] | Motion correction | $F_A, S_A$ | $\vert$ | – | Qualitative | Custom SD-OCT | – |
|  | Chen [73] | Motion correction | $F_A, S_A$ | $\vert$ | Enhanc. Layer Seg. | Qualitative, Quantitative | C | 15– |
|  | Zheng [74] | Layer segmentation | $S_B$ | $\vert$ | Layer Seg. | Quantitative | E | 5– |
|  | Hendargo [75] | Motion correction, OCT montage | $C$ | $\parallel$, $\perp$ | Enhanc. Layer Seg. Vessel Seg. | Qualitative | M | 1H |
|  | LaRocca [76] | Handheld SLO-OCT device | $C, F_B, S_B$ | $\perp$; SLO | Enhanc. Layer Seg. Vessel Seg. | Qualitative, Multimodal | N | – |

**Table 1.** *Cont.*

| Year | Reference | Research Topic | Handled Plane | Method | Preproc. | Validation | OCT Type | Dataset |
|------|-----------|----------------|---------------|--------|----------|------------|----------|---------|
| 2014 | Chen [77] | Motion correction | $F_A, S_A$ | \| | Enhanc. Layer Seg. | Qualitative, Quantitative | C | 26P, 19H |
| | Kraus [78] | Motion correction | $C, F_A, S_A$ | $\perp$ | Enhanc. Layer Seg. | Qualitative, Quantitative | D | 73− |
| | Montuoro [79] | Motion correction | $F_A, S_A, F_B$ | \| | Layer Seg. | Qualitative, Simulated | B, E | 100 scans |
| | Wu [49] | Motion correction | $C$ | \| | Enhanc. Vessel Seg. RPE Seg. | Qualitative, Quantitative | SD-OCT | 15P |
| 2015 | Lee [80] | OCT surface registration | $F_B, C$ | \|\| | Layer Seg. | Qualitative, Simulated | O | 3P, 3H |
| 2016 | Lang [81] | Motion correction, OCT registration | $C$ | \|\| | Enhanc. Vessel Seg. Layer Seg. | Qualitative, Quantitative | SD-OCT | 39H |
| | Lezama [82] | Motion correction, OCT registration | $C, F_B, S_B$ | $\perp$ | Enhanc. Layer Seg. | Quantitative | P | 12− |
| | Cheng [50] | Motion correction, Multimodal registration | $C$ | \|; FP | − | Multimodal, Quantitative | F | 18− scans |
| | Fu [83] | Motion correction | $F_B$ | \| | − | Qualitative, Simulated | F | 7− |
| 2017 | Chen [84] | Motion correction, Scanning protocol | $C, F_A, S_A$ | \|\|; SLO | Enhanc. | Qualitative, Multimodal | Q | 1− |

**Table 2.** Specifications summary of the OCT devices used in the reviewed procedures. Variables described as TD: Time domain, SD: Spectral domain, SS: Swept-source, and $\times$: Not included or mentioned.

| Code | Device | OCT Type | Acquisition Rate (A-Scan s$^{-1}$) | Axial Resolution in Air (µm) |
|------|--------|----------|-----------------------------------|------------------------------|
| A | Stratus OCT3 | TD | 400 | 10 |
| B | Cirrus HD-OCT | SD | 27k | 5 |
| C | Spectralis OCT | SD | 40k | 7 |
| D | RTVue | SD | 26k | 6 |
| E | Topcon SD-OCT1000 | SD | 18k | 5 |
| F | Topcon DRI OCT-1 | SS | 100k | 8 |
| G | Custom | TD | 40 | 14 |
| H | Custom | SD | 18k | 4.5 (retina) |
| I | Custom | SD | 70–312.5k | 3.6–11.6 |
| J | Custom | SD | 20k | $\times$ |
| K | Custom | SD | 70k | 3 |
| L | Custom | SS | 200k | 7 |
| M | Custom | SS | 100k | $\times$ |
| N | Custom | SD | 20k | 7 |
| O | Custom | SS | 100k | 2.7 |
| P | Custom | SS | 100k | 4.7 |
| Q | Custom | SS | 100k | 8.5 |

### 4.1. [1991–2005]

Although OCT devices date from 1991 [1], the first publication on measurements of human retinal structure in vivo was published in 1993 [85]. During these early years, when TD-OCT was used, only axial misalignments between adjacent A-scans (fast axis) were corrected using cross-correlation. The registration was considered as part of the pre-processing, and no information was given about the validation or the computation time.

**Swanson et al.** [53] **(1993)** and **Schuman et al.** [54] **(1996)** obtained an estimate of the retinal motion using a cross-correlation technique in which the axial index was chosen as the location of the peak correlation of adjacent A-scans. Cross-correlation was also used by **Ishikawa et al.** [55] **(2005)** to improve macular segmentation for glaucoma assessment. The A-scan lines were shifted so that the sum of the products of adjacent pixels could be maximized. Forty-seven subjects (23 normal and 24 with glaucoma) were analyzed in this study.

### 4.2. [2006–2008]

The establishment of the SD-OCT allowed to acquire faster and larger OCT volumes, which required the introduction of new techniques for correcting misalignments between cross-sectional images (slow axis), as shown in Figure 6.

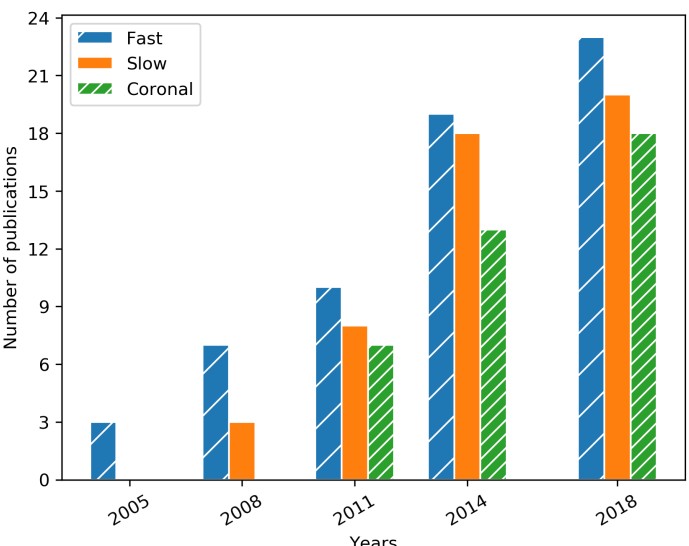

**Figure 6.** Cumulative bar plot showing the number of reviewed publications that included OCT retinal motion correction handling the fast, slow, and coronal planes.

**Zawadzki et al.** [19] **(2007)** proposed new methods for correcting eye motion and scanning beam distortions. For the motion artefacts, a rigid pairwise registration (only translation) based on auto-correlation was used to correct distortions between consecutive B-scans. The concept of complementary data appears here for the first time, as additional B-scans were acquired along the slow axis (with vertical scanners scanning fast) and used as a reference for axial shifts of consecutive B-scans. The procedure was tested in two volumes of the same subject, and validated with visual inspection.

### 4.3. [2009–2011]

Up to this stage, the retinal motion correction was tackled without using additional imaging modalities. This changed after 2008, when more approaches started to include other modalities such as SLO or FP to support the data visualization and registration correction, as shown in Figure 7.

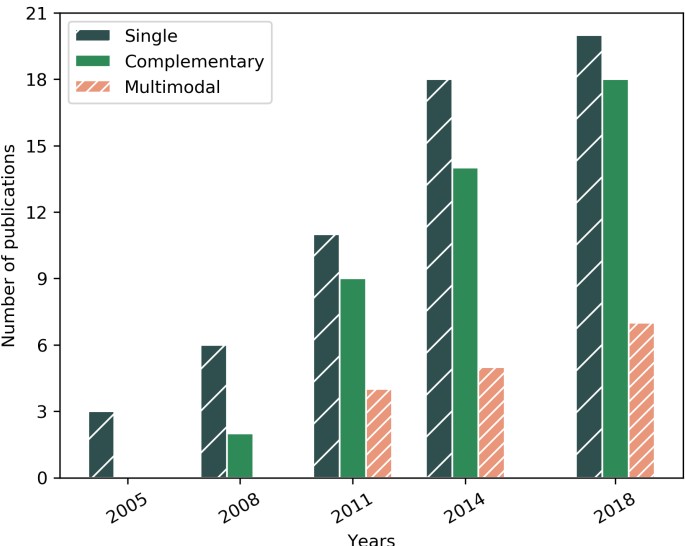

**Figure 7.** Cumulative bar plot showing the number of reviewed publications that used OCT single, OCT complementary, and multimodal acquisitions for correcting the retinal motion.

**Ricco et al. [59] (2009)** proposed a method to correct coronal misalignments by registering the OCT volume to an SLO image. This study included volumetric scans of 48 healthy and 116 glaucomatous eyes. As pre-processing, a vessel segmentation algorithm proposed by Lam and Yan [86] was used. The registration procedure had two main steps. The first step used a non-rigid technique proposed by Periaswamy and Farid [87], where motion between two images is described by an affine transformation. The second step corrects vessel discontinuities by finding the horizontal shift at each pixel in the en face OCT image that best aligns the result with the SLO image [88]. The validation was done qualitatively by visual inspection and quantitatively by computing the improvement in overlap between vessels on the SLO image and on the motion corrected en face image.

**Niemeijer et al. [60] (2009)** proposed a feature-based method to register OCT volumes. It was tested on 12 SD-OCT scans of six normal volunteers. As pre-processing, retinal layers of each volume were segmented and flattened. Then, a vector of thousands of 2D Scale-Invariant Feature Transform (SIFT) features [89] was generated on each image, and the correspondence between points belonging to two volumes of the same patient was established by comparing the distances. Only translation and rotation transformations were considered, and the distance between features was used to guide the registration. The validation was done qualitatively by visual inspection of the vessel pattern on the OCT en face images, and quantitatively by simulating deformations applying scaling, and rotation to the 3D-OCT scans.

**Xu et al. [62] (2009)** proposed an automated system for eye motion correction in SD-OCT volumetric data. They used a shape context algorithm that aimed at preserving the natural contours of the optic disc. Forty-three eyes (19 healthy and 24 glaucomatous) of 39 subjects were enrolled in this study. The projected en face OCT data was registered to the corresponding SLO image using a global transformation, computed based on retinal blood vessel maps detected on both SLO and OCT. Visual inspection was performed by three different observers.

*4.4. [2012–2014]*

By the end of 2011, the attention given to the pre-processing started to grow as more complex algorithms for OCT data registration were developed over the following years, as depicted in Figure 8.

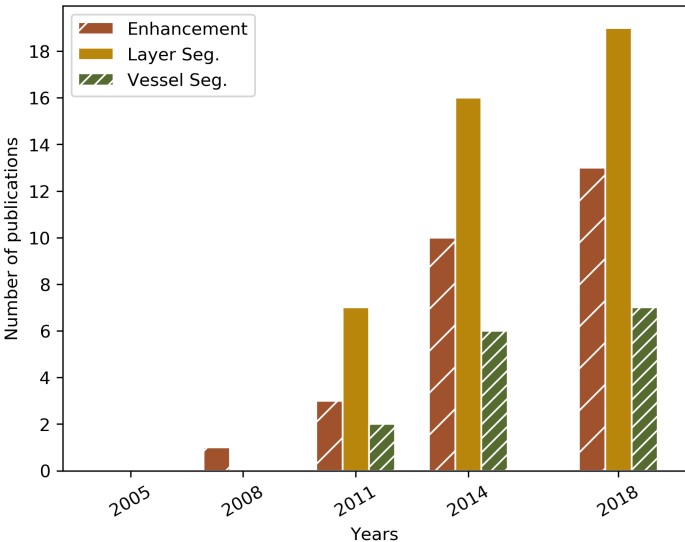

**Figure 8.** Cumulative bar plot showing the number of reviewed publications that applied enhancement, layer segmentation, and vessel segmentation as pre-processing in the registration correction.

**Chen et al. [73] (2013)** proposed a new deformable algorithm for intra-volume OCT registration based on the similarity between pairs of A-scans. All the images were first pre-processed by removing the background above the ILM and below the Bruch's membrane (BM) boundaries. The registration consists of three steps. First, the B-scans were registered using only translations in order to align data in the fovea. Next, using the retinal boundaries as landmarks, an affine transformation (translation and scaling) was applied to each pair of A-scans (from the reference and target images). Finally, a deformable registration that uses regularized one-dimensional radial basis functions [90] was applied, computing the sum of squared differences (SSD) between A-scans as the similarity measure.

Fifteen volumes with manual segmentations of eight retinal layers were used to evaluate the registration performance. Two of the images were randomly chosen as atlases and registered to the other 13 images. With the estimated deformation field, atlas labels were transformed to each target space and the Dice similarity coefficient between each of the eight layers was reported. **Chen et al. [77] (2014)** followed the previous conference publication and extended the work in a journal paper. A larger validation cohort consisting of OCT images of the right eyes from 45 subjects (19 healthy controls and 26 patients diagnosed with multiple sclerosis) was used. Intensity normalization using a linear contrast stretching and the segmentation of another extra layer (nine in total) were considered in the pre-processing.

**Hendargo et al. [75] (2013)** proposed a method to eliminate motion artifacts by creating a composite image. Multiple OCT volumes from a healthy 27-year-old volunteer were acquired sequentially. The method consisted in stitching together motion-free patches from each dataset. This approach also applied segmentation of the retinal layers, and used the vascular network on each layer to guide the registration. To ensure that the registration used only vascular information, binary masks were obtained. Similarly to [70,74], it also proposes a two-step global-to-local approach. First, the FFT and correlation are used to register individual stripes from different volumes in a rigid manner. Then, spline-based registration was used to locally deform the vessels to show a continuous, smooth transition between different stripes. Visual inspection was used as validation technique.

**Kraus et al. [78] (2014)** proposed an improvement of their registration algorithm [71]. They stated that, given the number of alterations that can affect the A-scans, such as vignetting, floaters or lens opacity, corresponding anatomical locations in two input volumes might have very different signal levels and A-scan signatures. Hence, in pre-processing, a bias field estimation based approach to correct illumination differences in the input volumes was employed. An intensity similarity measure using the pseudo Huber norm and a regularization scheme based on a pseudo $L_{0.5}$ norm was introduced.

Similarly to [71] two orthogonal volumes were scanned, and the two-step approach for the registration was developed. In the first stage, only axial motion and axial tilt are coarsely corrected, assuming that the B-scans were rigid structures that can move only on the axial direction. This result was then used as the starting point for a second stage full optimization. Quantitative evaluation was performed using data acquired from 73 healthy and glaucomatous eyes using SD-OCT systems. The algorithm was evaluated measuring retinal thickness, and comparing the results with the previous approach [71] and with the volumes without registration. In addition, the repeatability was evaluated, comparing several independent OCT volumes of the same subject by means of global measures, such as MI of the volumes, and local ones, such as comparison of vessels and layers.

**Montuoro et al.** [79] **(2014)** proposed a method to correct motion artefacts using a single volume scan while still retaining the overall shape of the retina. It is stated that a volume acquired from a healthy individual should be mostly locally symmetric along the axial scan direction (coronal plane). Thus, the appearance of a small window in the fast scan direction should be similar to the appearance in the slow scan direction. In this study, a set of 100 macula centred SD-OCT volumes of two diferent SD-OCT scanners from patients with central retinal vein occlusion were used. The estimate of the local retina curvature by a preliminary segmentation of the RPE was obtained as a pre-registration step. Hence, a window in the fast and slow scan directions were selected and the local curvature was obtained by performing linear regression using the least-squares method. The axial correction was performed by shifting each A-scan along the depth axis based on the minimum sum of squared distances between the local and fitted curvature. In order to compensate for lateral motions (fast axis), local pairwise phase correlation between B-scans was calculated. The method was qualitatively validated by a group of trained OCT grading experts on 100 SD-OCT scans and quantitatively validated using a set of synthetic SD-OCT volumes. Furthermore, the motion compensation estimation by the proposed method was compared with a hardware eye tracker on 100 SD-OCT volumes.

**Wu et al.** [49] **(2014)** proposed a registration method for pathological retinal 3D SD-OCT volumes based on Coherent Point Drift (CPD) [91]. Pre-processing included flattening of the B-scans to accurately locate the RPE layer and obtain its projection, denoising using a block matching based sparse transform [92], removal of non-vessel shadows, and vessel segmentation including its shadows. The motion correction algorithm defined two point sets, source and target 3D vessel center lines. One of the sets represents the Gaussian mixture model (GMM) centroids, and the other represents the data points. The sets are aligned solving a probability density estimation problem. Affine and non-rigid registrations were examined here. The optimization for affine was unconstrained, whereas, for non-rigid, a regularization of the norm of the displacement function [91] based on the motion coherence theory [93] was used. To evaluate registration performance, qualitative analysis of the transformed point set bifurcations by experienced graders was performed. A better result was observed of the non-rigid transformations. In addition, a total of 15 unique 3D SD-OCT fovea-centred diseased volumes were used for testing registration performance by calculating the modified Hausdorff and Mahalanobis distances [94] between point sets prior and post registration. The point set distance was expected to decrease post registration as the retinal vessels become aligned.

*4.5. [2015–2018]*

Although the first registration manuscript using SS-OCT mentioned in this bibliographic review appears in 2012 [71], these devices were not widely used until 2015. By then, an A-scan acquisition rate also raised up to 100 kHz, contributing to a reduction of the motion artefacts observed in the fast plane.

**Lang et al.** [81] **(2016)** proposed a method to jointly do registration and motion correction between longitudinal data. A dataset of 26 healthy control subjects was used in this study. The method required complementary data (volumes acquired from the same subject at different times). A fundus projection image (i.e., an image obtained by averaging the intensity in the depth direction between certain layers) was created using each volume. Then, a set of points representing the vessels were extracted from these images. The correspondences between the sets of points were estimated using CPD [91].

CPD introduces non-rigid deformations, and the authors aimed to restrict the deformations to rigid plus a scaling component. Thus, a cost function was minimized, iterating between the point-based registration and a lasso regression. The optimization of this cost function was done interpreting it as a simple least squares point-based registration problem. Visual evaluation and average RMSE of the manually selected landmark points after registration were used as validation techniques.

**Lezama et al. [82] (2016)** proposed an algorithm that combined segmentation and registration using orthogonal volumes. As pre-processing, retinal boundaries in each B-scan were segmented and used to produce an enhanced en face image of the retina. These en face projections were used for lateral registration in three steps. First, the saccadic eye motion was detected, and the saccade locations were used to split the volumes. This was done based on discontinuities in the en face projection detected by lower correlation with the surrounding regions. Second, the inter-saccadic intervals were stitched using orthogonal scans as reference and a locally affine model. Third, an orientation-aware optical flow algorithm for dense local registration was applied. This results in motion only in the fast-scan direction. Once the affine transformations and local displacements had been found for each saccade-free region of the enhanced en face image, they were applied to the A-scans in order to correct the (lateral) motion in the whole 3D volume. In the last stage of the algorithm, the segmented retinal surfaces of each volume were used to correct for motion in the axial direction (vertical shifts of the A-scans). For validation, the MI, RMSE and distance between layers were computed for the retinal volumes of 12 subjects. The differences between vascular trees and artificial data with known movements were also used for validation.

**Cheng et al. [50] (2016)** proposed a novel method to correct the misalignment between consecutive B-scans to improve the accuracy in multi-modality retinal image registration. The method was motivated by the fact that pixels from static tissue yield small decorrelation values whereas pixels from the faster moving tissue yield high decorrelation values. Hence, decorrelations from overlapping scans were computed and used to create an en face decorrelation image. The B-scans affected by motion were aligned (translation) with their precedents using a iterative diamond search strategy [95]. Lastly, the corrected OCT en face image was registered to a color fundus image using a matching method based on the low-dimensional step pattern analysis [96]. Here, 28 pre-defined features were matched by Euclidean distance. Then, a random sample consensus [97] was used with an affine transformation to produce the final registration. In order to validate, the RMSE, the maximum absolute error, and the median error of 18 3D-OCT scans and the corresponding color fundus images were used.

**Fu et al. [83] (2016)** proposed a pairwise registration approach based on saliency and center bias constraints. The saliency detection was used to separate the layers region and the background. The transformation model was estimated based on the higher weight on the saliency region, with different weights given to background and foreground. With the saliency map, the transformation model was estimated for each OCT slice based on the pixel intensity and gradient value. A center bias constraint was also included. Here, a Gaussian-based function was used to encourage that a drifted slice remained close to its initial position if the amount of drift in the fast axis was low. Lastly, transformation vectors of the objective function were obtained by considering two terms related to feature similarity in saliency and center bias constraint. In order to validate the proposed algorithm, seven OCT volumes were used to generate artefacts in 49 synthetic OCT volumes. Visual inspection of the blood vessel discontinuity was also considered.

**Chen et al. [84] (2017)** proposed an imaging method based on Lissajous scanning and a respective motion correction algorithm. Lissajous scan is a closed-loop scan obtained by operating along the horizontal and vertical axes using sinusoidal waveforms of different frequencies. The core idea of the algorithm proposed in this work was to correct motion artefacts by maximizing the similarity of overlapping areas between multiple scans. The first step of the algorithm (pre-processing) was to split the OCT measurement data into smaller sub-volumes. The correction process was divided into two parts, lateral motion correction and subsequent axial motion correction. For the lateral motion correction, the en face projection of each sub-volume was used. Here, a rough and a fine correction

were applied based on the maximum of the cross-correlation function. For the axial motion, the 1D cross-correlation was calculated in the overlapping area between the reference and registered A-lines. The axial shifts of the registering A-lines were determined by the maximum peak positions of the cross-correlation function. For validation, they compared the motion corrected results with SLO and cross-sectional OCT along fast scan direction as reference standards. To validate the repeatability, two motion-corrected volumes, which were measured in the same location using the same subject but at different times were compared. In addition, visual inspection was used. This algorithm was later extended to OCT-A [98].

## 5. Discussion

Retinal imaging has shown a growing potential in the early diagnosis of pathologies that are related to the physiology and inner structure of the retina and its adjacent tissues [99–101]. The rapid development of better imaging devices, as well as the research in image processing to enhance and correct the acquired data, will certainly lead to the development of new retinal biomarkers. In the upcoming years, these biomarkers will aid the diagnosis and prognosis of not only ophthalmic, but also cardiovascular and neurodegenerative diseases [102,103]. As shown in this review, one of the most important research lines on retinal image processing is the motion correction, as it is relevant not only for the correct display and visualization of the volumetric data, but also for the reliability of retinal imaging biomarkers.

Motion correction approaches can be roughly divided in prospective and retrospective. On one hand, prospective techniques use specific hardware to collect additional eye motion information during the acquisition of the OCT data. This is probably the most common motion correction approach in the clinic, since many commercial systems use SLO or fundus images to correct the OCT data. A number of studies have also presented specific prospective techniques to correct retinal motion. For instance, Pircher et al. [14] used an SLO simultaneously with OCT imaging in combination with axial motion correction. Axial motion was corrected measuring the position of the corneal apex using an additional spectral domain partial coherence interferometer integrated into a transverse scanning OCT system. With this information, the reference arm length of the imaging interferometer was adapted to correct for axial motion and allowed for axial tracking with an accuracy below the axial resolution of the system [104]. Sugita et al. [41] performed transverse motion tracking by correcting the actuations of the horizontal/vertical-scanner in a polarization sensitive OCT system. This was done according to the detected retinal position shifts measured by a simultaneously operated line-scanning laser ophthalmoscope. Using the cross-correlation values of the parallel acquisition, the best match was found, and the horizontal/vertical shifts were obtained. A correction signal proportional to the computed shift information was then sent to the *XY*-scanner of the OCT. Tracking rates up to 60 Hz were achieved with this system. However, the introduction of AO-OCT requires a more demanding retinal stabilization and, consequently, further research on prospective approaches. Kocaoglu et al. [105] presented a customized retinal tracking module integrated into the sample arm of an OCT system, able to correct eye movements up to 100 Hz, and hence reduce residual motion to 10 μm root mean square. Despite the in situ correction, prospective approaches may cause additional strain to subjects with fixation limitations, as they have to go through a longer exposure. For example, an instrument with online correction stops recording if a misalignment is detected, leading to a longer exposition on subjects with fixation problems. However, in the absence of fixation problems, such as in healthy individuals, one could argue that this solution is appropriate for correcting retinal motion, as it is done online instead of as part of the post-processing. Nevertheless, eye trackers do not remove the need of using motion-correction algorithms [106]. Prospective techniques also require specific additional hardware, which must be customized to a number of applications, increasing the complexity of the imaging device. On the other hand, retrospective techniques, which are the focus of this review, perform a correction based on assumptions about the retinal motion patterns. It is an offline procedure that may include additional references and, unlike prospective approaches, it can be applied to OCT



data acquired previously, allowing its inclusion in longitudinal studies. Retrospective procedures may involve single, multimodal, or complementary acquisitions. Multimodal acquisition refers to offline alignment of two or more imaging modalities. In complementary acquisitions, different protocols are used to acquire data at the same location. Single acquisition includes the corrections implemented when only one image or volume is available.

An optimal procedure for correcting OCT data misalignment could be achieved including information on the underlying retinal motion model, that is, to use theoretical representations that integrate knowledge on the frequency, amplitude and direction of the retinal motion. Unfortunately, there are a few drawbacks that limit the application of this procedure in OCT imaging. First, it is necessary to include information about the acquisition settings. This might increase the model complexity substantially, as many configurations are possible (e.g., field of view, axial and lateral resolutions, acquisition rate, scan pattern, etc). In addition, unlike other types of body motion (e.g., respiratory, cardiac motion), retinal motion results from saccades and drifts, which vary on amplitude, direction, and frequency, making its combination difficult to predict. Moreover, retinal motion may differ significantly between individuals [12], hindering the development of a general model. Thus, the correction based on theoretical modelling has been on hold, although the continuous collection of large datasets using retinal trackers may allow the use of advanced learning models to predict the retinal motion in the near future.

Currently, coronal and axial misalignments due to involuntary eye movements remain a major problem on OCT imaging [13,14], affecting both its quality and reliability. Subtle structural changes on the retinal tissue due to the early manifestation of a pathology can be easily understood as a misregistration problem. Pre-processing techniques can be used to mitigate these issues. Even if a common approach does not exist, noise removal using filtering (e.g., median, Wiener, Gabor), downsampling to reduce the computational complexity, and normalization, are often used. Depending on the application, to take into account specific areas of interest can also be helpful to enhance the registration performance. Background removal, retinal layer segmentation, or vasculature segmentation have been reported in a number of studies so far. The background removal is important in those cases where the SNR is relatively low and may have a significant weight on the registration performance. Retinal layer segmentation has been extensively used to guide image registration in OCT volumes, specifically if their position and shape are of high interest for the research problem. However, a precise layer segmentation is far from a straightforward task, so most of the publications opted for segmenting only a few layers, usually the retinal boundaries and the RPE layer, as they have the highest reflectivity, making them easier to distinguish. A step commonly associated with layer segmentation is the flattening, which facilitates the segmentation by removing the retinal curvature. However, depending on the application, this curvature has to be recovered later. Main vessel segmentation is also often used, specifically if multi-modality data are available, such as OCT en face imaging and SLO or FP. In order to facilitate the vascular segmentation, it is common to apply spatial filters, such as Gabor or Laplacian of Gaussian.

The first publication in OCT image registration dates to 1993 and, until 2005, only axial misalignments were taken into account. The first devices only captured a few B-scans, so the correction was only done within a cross-sectional image. Cross-correlation was by then the most popular metric, although it may not be sensitive enough to discriminate between the smallest structural changes among neighbouring OCT scans [33]. The fast image acquisition introduced by SD-OCT allowed for obtaining several B-scans conforming a volume. This motivated the development of new methods to correct axial misalignments seen across the slow axis of a volume scan. The cross-correlation metric remained a popular choice, and was used in combination with pairwise rigid and, later in 2008, affine transformations. The need of a reference led to the development of multimodality imaging, broadly raising the interest on feature-based registration. Rigid and non-rigid transformations were used to align common features between OCT and SLO or FP. Although it is assumed that SLO can be used as a ground truth, it is not completely free from movement artefacts due to its scanning pattern [64]. FP can

be considered a better reference since the whole image is taken in a single shot. The introduction of retinal multimodality imaging allowed, for the first time, to have a reference for coronal misalignments. By the end of 2010, despite all the progress done, axial (fast and slow axis) misalignments remained a significant problem on OCT image reliability. Hence, changes on the acquisition protocols were suggested as a mean to obtain a reference for the registration.

Based on the artefact difference between the slow and the fast-axis, a second volume, orthogonal to the first one, could be used to correct the axial misaligments over the slow axis in the main (reference) volume. Feature-based registration, focused on both the retinal layers and the blood vessel ridges, was used to correct these misaligments. The establishment of SD-OCT as a gold standard imaging modality in ophthalmology boosted its development to SS-OCT. The joint efforts of the scientific community on increasing the acquisition rate led to cross-sectional images that are nowadays being considered movement artefact free. Consequently, since 2015, the correction of the misalignments between A-scans handling the fast axis has significantly dropped, and most authors dedicated their efforts to correct between B-scans on both axis, between A-scans across the slow axis, and using the coronal plane.

Moreover, these last years have also been marked by the wide adoption of OCT-A imaging, where small motion artifacts have shown a large impact on the angiographic image quality. Some methods have been proposed to correct motion artefacts in OCT-A en face imaging. For example, Zang et al. [107] acquired two non-orthogonal en face angiograms and divided them into microsaccade-free parallel strips. Then, a rough registration based on large vessels and a fine registration based on small vessels were sequentially applied to register parallel strips into a composite image. Although some correction methods are based on the OCT-A en face imaging, the majority of the motion correction algorithms employ the techniques mentioned in this survey, which were developed and tested for correcting intrinsic retinal motion in conventional OCT imaging. For instance, the methodology proposed by Kraus et al. in 2012 [71] and later in 2014 [78] has been widely used as a standard correction in OCT-A imaging [13,108–113]. The fast rise of OCT-A as a non-invasive vascular imaging modality in clinical practice has brought up again the importance of proper and accurate registration in OCT imaging. Since OCT-A imaging results from the amplitude decorrelation or phase variance of two or more OCT images, it is fundamental to ensure that the OCT data are free of movement artefacts in order to provide reliable information of the retinal systemic system. The need of correcting the OCT-A images shows that, despite all the advances in motion correction since the introduction of OCT, more knowledge in the intrinsic retinal motion process, and more advanced and precise methods to correct these artefacts in OCT imaging are needed. Moreover, although the combination of a high acquisition speed, multi-imaging protocols for complementary data, and multi-modality, are reducing the retinal imaging artefacts to extremely low levels, the increase of the lateral and axial resolution is expected to bring new challenges to image registration. In addition, the performance of the registration is not clear when pathological data are analysed, emphasizing the importance of proper validation techniques.

Unlike other imaging technologies, retinal OCT data do not have a ground truth reference to validate the registration. Therefore, qualitative techniques and, specifically, visual inspection, have been the main validation techniques used to infer the performance of the retrospective registration in OCT imaging. The low acquisition rate of the first OCT generation, TD-OCT, required to inspect the misalignments within each cross-sectional image. Since the introduction of SD-OCT, B-scans are obtained with acquisition rates comparable to the frequency of the drifts or microsaccades. Hence, both registration and validation have shifted towards the correction of misalignments handling the slow-axis (between adjacent B-scans). A smooth profile, similar to what can be observed in the fast-axis, is expected over the slow-axis. However, retinal pathologies such as age-related macular degeneration (AMD) or macular hole are likely to result in steep adjacent regions, which may be wrongly understood as consequence of a poor registration. In order to distinguish if the lack of smoothness is caused by motion or pathologies and, hence, if it must be corrected or not, the registration started to include the acquisition of additional orthogonal B-scans along the slow axis of the main image. Some authors

used these B-scans for validation, as a reference of the expected retinal profile on the volumetric OCT data. This provides a rough idea of the axial registration performance over the fast and slow planes, but not on the coronal plane. For the latter, en face images obtained by projecting the volumetric data can be used. The visualization of structures, such as the continuity of the blood vessels and the shapes of the optic disc or the fovea, gives hints on the performance of the registration algorithm. An easy way to infer the registration robustness through visual inspection is to use a checkerboard between two independent measurements of the same subject, either between cross-sectional or en face images.

Although visual inspection gives a first insight on the registration accuracy, it is insufficient for evaluating the performance of registration algorithms because there is neither a standard reference nor a ground truth available. Additionally, it does not permit the optimization of an algorithm (e.g., fine-tuning registration parameters). Hence, as the dimension of volumetric datasets started to increase with the switch from TD-OCT to SD-OCT, quantitative validation techniques became an urgent need. Consequently, since 2007, a number of methods have been proposed, which have been categorized in this review as quantitative metrics (intensity-based or comparison of annotated structures), simulated data, and multi-modality comparison. One of the most widely used intensity-based techniques in the field of OCT registration is mutual information, which compares the intensity distribution of the histograms of the images. Regarding the comparison of annotated structures, two main groups can be distinguished: to compare the overlapping of volumes or surfaces and to compute the distances between a set of landmarks. In the first group, the most common formulation in the works that were reviewed is the Dice similarity coefficient. In the second group, there is a wider variety of metrics, as different distances have been used. One common application is to compare an OCT volume before and after registration, marking the same points (e.g., vessel bifurcations) in the B-scans from the original and registered volume. The distance between the same point in different B-scans is expected to decrease post registration, as the retinal structures become aligned.

The metrics based on annotated structures can be computed using a manually labelled reference or a fully automatic approach. The first provides a ground truth but has the main drawbacks of being subjective and time-consuming. The latter depends on the quality of the automatically computed landmarks or regions, which can bias the results, but it is deterministic and faster. The unique morphology of the retinal shape or the expected thickness of the retinal layers have also been explored as retinal landmarks to validate the registration techniques. Techniques based on annotated structures are suitable to validate inter-volume registration. However, special care must be exercised when analysing intra-volume misaligments, as significant changes between B-scans are expected even within accurately registered data. These changes take place mainly between steep adjacent areas (e.g., optic disc) or if a pathology is present (e.g., macular hole). Known deformations in simulated data have also been used for validation purposes. One option is to apply specific transformations (e.g., translation, shearing, rotation) and verify if the registration recovers them accurately. However, to cover all the possible misregistrations is a challenging task, as it is complicated to accurately model retinal motion. Some authors proposed to learn these misregistrations from an unregistered OCT dataset, but this requires a large amount of data to ensure that the artefacts are representative. Finally, additional modalities that are considered motion-free, such as FP or SLO, can be used as a ground truth to correct retinal motion in OCT. A special case for validation would be the retinal trackers, which perform an online correction to improve the quality of the images during the acquisition.

The goal of the motion correction in OCT volumes is not only to ensure that the images are accurately aligned, but also to understand the nature behind the motion. Analyzing the advantages and drawbacks of the approaches studied in this review, it can be concluded that the best approaches are those that combine the orthogonal volume acquisition, as they are able to correct the axial misalignments over the slow and the fast axis, with the multi-modality acquisition, as they can correct coronal misalignments. However, an approach that uses these techniques is less convenient from a practical point of view due to its higher complexity and longer acquisition time. An algorithm that requires only one OCT volume would be simpler, with the additional advantage of allowing the processing

of previously acquired volumes. Most of the registration procedures used a pairwise approach. A common problem is the incorrect registration between B-scans that have not been affected by retinal motion. One possible solution is to include constraints that restrict the type and/or amount of the transformations. Even though non-rigid transformations may provide some flexibility to the registration, it can also cause severe deformations in the retinal structure that can be easily missed during the visual inspection. Therefore, as these algorithms do not use references, a strong validation is required to assess their performance.

Some publicly-available registration frameworks have been made available to facilitate the research in medical image registration [114] as well as the comparison of different registration procedures. In general, the largest collection of image registration tools can be found in the Insight ToolKit (ITK) [115]. Other works, using and extending ITK, are Advanced Normalization Tools (ANTs) [44,116,117], elastix [43], plastimatch [118,119], and Deformable registration via attribute matching and mutual-saliency weighting (DRAMMS) [120]. Moreover, there are a number of non-ITK-based implementations such as NiftyReg [121]. These frameworks include several optimization methods, multiresolution schemes, interpolators, transformation models, and cost functions, allowing for studying the effect that each component has in the registered image. These image registration tools may be a good starting point for those aiming to do retrospective OCT motion correction.

## 6. Conclusions

In this work, an extended review on retrospective registration used for OCT retinal motion correction is provided. We show how the increase in imaging acquisition rate, from the first TD-OCT systems at a few A-scans/second, to the current SS-OCT devices at a few millions A-scan/second, has changed the main motivation behind the retrospective motion correction over the past few years. The nature of the motion artefacts is discussed as well as the registration techniques. The pre-processing and the current validation methodologies are also detailed. Despite the simplicity of the retrospective approaches compared to hardware solutions, their validation remains a major concern and limitation, compromising the reliability of the registered data, specifically on pathological images. The development of faster acquisition devices will certainly contribute to reduce the retinal motion artefacts in SS-OCT imaging. However, the integration of technologies such as AO-OCT and OCT-A sets even higher demands on image acquisition speeds. It has already been shown that registration algorithms will be fundamental to ensure a high in vivo retinal imaging quality with a cellular structure resolution. Besides drifts and microsaccades, retinal tremors will be a new variable to consider in the retinal motion correction models. Hence, new precise registration procedures incorporating different OCT modalities (OCT, AO-OCT, OCT-A) together with robust validation techniques should be tackled by the OCT scientific community over the upcoming years.

**Author Contributions:** Conceptualized and designed the review: L.S.B., D.A.D.J., T.v.W., and S.K.; Drafted the manuscript: L.S.B., and D.A.D.J.; Revised the manuscript for intellectual content: L.S.B., D.A.D.J., M.P., M.F.S., T.v.W., and S.K.

**Funding:** This work was supported by the Horizon 2020 research and innovation programme (Grant No. 780989: Multi-modal, multi-scale retinal imaging project).

**Conflicts of Interest:** The authors declare that there are no conflicts of interest related to this article.

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
