# Peer review of "Review on Retrospective Procedures to Correct Retinal Motion Artefacts in OCT Imaging"

_applsci, doi:10.3390/app9132700_

Round 1
Reviewer 1 Report
This manuscript reviewed the retrospective procedure to correct the retinal motion and axial eye motion artefacts in optical coherence tomography (OCT) imaging. Different techniques were described in this manuscript.
1. This article would have been more attractive if section 4 can be shorter. It’s not necessary to list each reference paper as a single paragraph.
Author Response
Response uploaded as PDF file

Reviewer 2 Report
Overall this is an important topic to review. Registration of OCT images can have drastic effects on clinical interpretation. This review has a good summary of the causes of motion artifacts and types of involuntary eye movements. This review excludes prospective and online approaches, which I think is a major weakness because many commercial systems use SLO or fundus images to ensure the OCT is artifact free. This is probably the most common motion correction method in the clinic. The structure of the methods reviews needs work. The article chronologically summarizes individual publications then discusses them separately. Describing each class of registration approach generally, while offering citations in support would be preferable. Also, this article is devoid of mathematical or algorithmic representations of the registration methods. The reader should have an idea of how to write an OCT registration program by reading this review. This review article covers a lot of information but would benefit from a drastic restructuring.
1. Figure 4 structure is strange. Probably better as a table or a flow chart.
2. Include mathematical descriptions of registration algorithms.
3. Include some description of prospective/online methods.
4. More rigorous descriptions of why registration is important for OCTA.
Author Response
Response uploaded as PDF file

Reviewer 3 Report
Review
This is a manuscript that discusses the state-of-the-art in the field of correction of retina motion artefacts in optical coherence tomography (OCT) imaging. While in vivo/in vitro microscopy via OCT is a powerful diagnostic methodology currently used in the best clinics worldwide, some methodological issues remain. Authors performed a very intensive literature review and included details of literature search.
While the manuscript in the present form is too long, I think the paper in a shorted form could prove to be interesting and useful to the audience, making it acceptable for publication in the Journal after a major revision.
Critiques:
Consistency:
state of the art (line 2) vs. state-of-the-art (line 27);
2kHz (line 31), 1050nm (line 105) vs. 20 um (line 116);
6's-1 – unit consistency (line 1123Fig).
Authors might consider making the manuscript shorter via removing parts that are not presented with sufficient details or duplicating the information.
While hardware approaches are well presented, only a few publications on software-based approaches were covered (line 52-58).
\Authors might consider removing literature search methodology (3.1 starting line 155) preserving only Fig.4 or removing both.
Authors might consider removing duplicating text. For example, volumetric OCT (line 70) was already introduced (line 21-22).
While lines 70-74 and Fig.1 are presenting the same information, authors might consider removing Fig. 1. Instead, authors might add the photo and description of the OCT system they used for imaging.
Authors might consider removing the basics of image processing (line 208-223).
Fig. 5 might be removed.
Line 338 – 341: Term FP was already introduced at line 257.Authors might consider re-writing the main part of the paper grouping similar works, instead of describing them individually.
Line 105: How many A-lines are within B-scan, what is the pixel size and estimated resolution?
Figures are missing gray-scale bars and scale bars are not readable.
Fig. 2: Consider adding a 3D view with imaging planes.
Fig. 3 and Fig. 7: Authors might consider adding some marks to the figure.
While feature-based registration is mainly combined with manual feature annotation, automated algorithms could be found in the literature demonstrating a good performance (line 233).
Fig. 6: 'Red' plane is specified in the annotation.
While time-domain OCT is used in the research application, it is almost not used in the clinical ones. Thus, the authors might consider removing it (line 270-273) as it is not directly related to the scope of the manuscripts.
Line 279: Authors might consider adding a reference.
Author Response
Response uploaded as PDF file
